# Effect of Vancomycin, Gentamicin and Clindamycin on Cartilage Cells In Vitro

**DOI:** 10.3390/biomedicines11123143

**Published:** 2023-11-25

**Authors:** Hermann O. Mayr, Nina Regenbrecht, Moritz Florian Mayr, Bianca Riedel, Melanie L. Hart, Hagen Schmal, Michael Seidenstuecker

**Affiliations:** 1Department of Orthopedics and Trauma Surgery, Medical Center-Albert-Ludwigs-University of Freiburg, Faculty of Medicine, Albert-Ludwigs-University of Freiburg, Hugstetter Straße 55, 79106 Freiburg, Germany; hermann.mayr@uniklinik-freiburg.de (H.O.M.); moritz.mayr@uniklinik-freiburg.de (M.F.M.); hagen.schmal@uniklinik-freiburg.de (H.S.); 2G.E.R.N. Tissue Replacement, Regeneration & Neogenesis, Department of Orthopedics and Trauma Surgery, Medical Center-Albert-Ludwigs-University of Freiburg, Faculty of Medicine, Albert-Ludwigs-University of Freiburg, Hugstetter Straße 55, 79106 Freiburg, Germany; nina.regenbrecht@uniklinik-freiburg.de (N.R.); bianca.riedel@uniklinik-freiburg.de (B.R.); melanie.lynn.hart@uniklinik-freiburg.de (M.L.H.)

**Keywords:** chondrotoxicity of antibiotics, infection prophylaxis, ligament surgery, concentration of antibiotics, LDH test, WST-1 test, Live Dead staining

## Abstract

Background: The treatment of grafts with vancomycin for ligament reconstruction in knee surgery is the current standard. However, high antibiotic concentrations have chondrotoxic effects. Purpose: To test the chondrotoxicity of clindamycin, gentamicin and vancomycin in comparable concentrations. In vitro and in vivo effective concentrations hugely vary from drug to drug. To allow for comparisons between these three commonly used antibiotics, the concentration ranges frequently used in orthopedic surgical settings were tested. Study Design: Controlled laboratory study. Methods: Human cartilage from 10 specimens was used to isolate chondrocytes. The chondrocytes were treated with clindamycin (1 mg/mL and 0.5 mg/mL), gentamicin (10 mg/mL and 5 mg/mL) or vancomycin (10 mg/mL and 5 mg/mL), at concentrations used for preoperative infection prophylaxis in ligament surgery. Observations were taken over a period of 7 days. A control of untreated chondrocytes was included. To test the chondrotoxicity, a lactate dehydrogenase (LDH) test and a water-soluble tetrazolium salt (WST-1) assay were performed on days 1, 3 and 7. In addition, microscopic examinations were performed after fluorescence staining of the cells at the same time intervals. Results: All samples showed a reasonable vitality of the cartilage cells after 72 h. However, clindamycin and gentamicin both showed higher chondrotoxicity in all investigations compared to vancomycin. After a period of 7 days, only chondrocytes treated with vancomycin showed reasonable vitality. Conclusions: The preoperative treatment of ligament grafts with vancomycin is the most reasonable method for infection prophylaxis, in accordance with the current study results regarding chondrotoxicity; however, clindamycin and gentamicin cover a wider anti-bacterial spectrum. Clinical Relevance: The prophylactic antibiotic treatment of ligament grafts at concentrations of 5 mg/mL or 10 mg/mL vancomycin is justifiable and reasonable. In specific cases, even the use of gentamicin and clindamycin is appropriate.

## 1. Introduction

Transplant-induced joint infection remains a severe complication in sports trauma due to its profound consequences, such as joint destruction and sepsis. The incidence of acute arthritis in cruciate ligament reconstruction with a tendon graft remains between 0.14 and 1.7% [1]. To prevent infections induced by the graft in ligament replacement surgery, irrigation of the graft and treatment with antibiotics have been established.

Surgical site infections (SSIs) can be prevented through the systemic administration of antibiotics during surgery. The effectiveness of systemic antibiotic treatment in orthopaedic surgery has been well-established, even though it has severe potential side effects [2,3,4]. Moreover, other methods of reducing the risk of infection are also being developed, such as local antibiotic prophylaxis [5,6]. Another promising strategy to prevent graft-induced infection is the use of a topical vancomycin application. Several studies have shown that the use of topical vancomycin application during spinal surgery has reduced the risk of infections [7].

Topical vancomycin is also increasingly used in orthopaedics. Tendon grafts for ligament reconstruction are often treated by an antibiotic solution in sports medicine to minimize the risk of infection. This involves treating the grafts with an undiluted vancomycin solution [8]. However, this method may considerably exceed the minimum inhibitory concentration (MIC). This value refers to the lowest concentration of an antimicrobial substance required to prevent the growth of microorganisms in a culture. The MIC is 0.06–0.12 µg/mL for clindamycin, 1–3 µg/mL for gentamicin, and 1–2 µg/mL for vancomycin [9,10,11].

The cytotoxicity of concentrated vancomycin (up to 50 mg/mL) was demonstrated in an in vitro study over 48 h [12]. Other studies have also demonstrated the cytotoxicity of various antibiotics against articular cartilage and other relevant cells [12,13,14,15]. Studies show that cytotoxicity is concentration-dependent [13,14,15].

The aim of the current study is to investigate the toxicity of clindamycin, gentamicin and vancomycin in human cartilage cells over the course of 7 days. The primary endpoint was the assessment of the relative survival rate of chondrocytes following a 168 h exposure to antibiotics at various concentrations, in comparison to a control group that did not receive antibiotic treatment.

## 2. Materials and Methods

### 2.1. Materials

Tissue culture vessels were purchased from Greiner Bio-One GmbH (Frickenhausen, Germany), Corning Incorporated (Kennebunk, ME, USA) and Carl Roth (Karlsruhe, Germany). DMEM + GlutaMAX and HAM’s F-12 Nutrient Mix (1x) (Art.No. 10565018) + GlutaMAX (Art.No. 61870036), which were used to prepare the specific chondrocyte culture medium, as well as phosphate buffer saline (PBS) (Art.No. 14080055), Penicillin-Streptomycin (Art.No. 15140148) and trypsin were obtained from Thermo Fisher Scientific (Karlsruhe, Germany), order number: 15400-054. Other reagents, such as fetal bovine serum (order number: S0615) (FBS), Amphotericin B (order number: P06-01100) and L-Ascorbic Acid Phosphate Magnesium Salt (order number: A8960), also used in culture medium, were purchased from Biochrom AG (Berlin, Germany) and Sigma Aldrich/Merck (Darmstadt, Germany). The enzymes Pronase (order number: 10165921001) and Clostridial Collagenase A (order number: 10103578001) were provided by Roche Molecular Biochemicals (Switzerland). The trypan blue dye (Art.No. CN76.3) was purchased from Carl Roth (Karlsruhe, Germany). Triton X-100 (Art.No. X100-100ML) for cell culture (as positive control of 100% apoptotic cells) was purchased by Sigma-Aldrich.

A 5810 R Eppendorf, (Hamburg, Germany) centrifuge was used. For cell counts and fluorescence microscopy, the light microscope Zeiss Primovert Microscope (Jena, Germany) and the Carl ZEISS Apotome.2 (Jena, Germany) were used. The incubator used to cultivate the cells was an Eppendorf Galaxy 170 R (Hamburg, Germany). For the ELISA measurements, a SPECTROstar^®^ Nano from BMG Labtech (Ottenberg, Germany) was utilised.

### 2.2. Methods

#### 2.2.1. Chondrocyte Isolation and Culture

Human cartilage was obtained from fresh tibial explants from hip and knee joints from ten different donors who underwent either posttraumatic- or degenerative-induced endoprosthetic replacement of the knee joint. While most of the articular surfaces of these joints were destroyed, the cartilage was taken from the remaining regions with undamaged cartilage. (mean age 61.7 years, six females, four males, age range 41–75; BMI 20–31). The study was performed using ten different donors. The samples from each individual donor were evaluated separately. The selected patients were informed about the procedure in advance and provided with a consent form to participate in the study. Furthermore, the experimental protocol was approved by the ethics committee of Freiburg University Medical Center (Ethics vote FREEZE 418/19). In each case, the native cartilage was first examined, and then the vitality of the cartilage was tested via the control group’s ability to undergo mitosis. The specimens were stored at 37 °C for 2 h until the harvested cartilage was minced using a disposable scalpel (PFM Medical, Köln, Germany) and washed with PBS. It was then placed into a 50 mL Falcon and covered in 10 mL of specific chondrocyte medium containing DMEM + GlutaMAX and HAM’s F-12 Nutrient Mix (1x) GlutaMAX at a ratio of 1:1, 2% Penicillin-Streptomycin, 1% Amphotericin B 10% FBS and 0.5% L-Ascorbic Acid Phosphate Magnesium Salt. The cartilage was left for digestion and contained 12 mg of Pronase, added through a PVDF sterile cell strainer. The samples were then incubated in an incubator at 37 °C, 95% O_2_ and 5% CO_2_. After 30 min, 12 mg of Clostridial collagenase A was filtered into the rudiment, again using a sterile PVDF cell strainer, and the solution was left at 37 °C, 95% O_2_ and 5% CO_2_. After 12 h, the solution was filtered through a 70 μm nylon EASYstrainer^TM^ and centrifuged at 1200 rpm for 10 min. The cell pellet was then washed three times with PBS, and the chondrocytes were suspended in the same specific chondrocyte medium as described above and cultured at 37 °C in 95% O_2_ and 5% CO_2_. 

#### 2.2.2. Chondrocyte Treatment

The cell medium was changed after 24 h and then every 48 h. After five days, the chondrocytes were washed with 5–7 mL of PBS. After washing, 3 mL of trypsin was added and the cells were left to incubate at 37 °C, 95% O_2_ and 5% CO_2_ for 5 min. A total of 7 mL of specific chondrocyte medium was then added, and the solution was centrifuged at 410 relative centrifugal force (rcf) for 14 min. The supernatant was discarded, and the pellet was resuspended in 1 mL chondrocyte medium. After staining, 10 µL of the 1 mL cell suspension was mixed with 10 µL trypan blue, and the cells were counted under microscope using a Neubauer Chamber Count. The chondrocytes were then seeded onto 48-well plates at a density of 40,000 cells/cm^2^ and cultured for 48 h to allow for attachment of the cells. After 48 h, clindamycin (Clindamycin Kabi 150 mg/mL, Fresenius Kabi Deutschland GmbH, Bad Homburg von der Hoehe, Germany), at the concentrations of 1 mg/mL and 0.5 mg/mL, gentamicin (Getamicin 80 Hexal^®^ SF 80 mg/2 mL, Hexal AG, Holzkirchen, Germany), at the concentrations of 10 mg/mL and 5 mg/mL, and vancomycin P (Vancomycin HEXAL^®^ 1,0 g, Hexal AG, Holzkirchen, Germany), at the concentrations of 10 mg/mL and 5 mg/mL, were added onto the chondrocytes and cultured for 24, 72 and 168 h following the addition of the antibiotics. Live Dead assay (containg calcein and propidium iodide) was obtained by biotrend (biotrend, Cologne, Germany). The concentrations of the antibiotics were chosen according to the literature in order to exceed the MIC and, if possible, to avoid a tissue-toxic concentration [8,9,10,11,12,13,15].

#### 2.2.3. Live Dead Staining

At the measurement times, 24, 72 and 168 h after antibiotic addition, the cells were washed and stained with calcein and propidium iodide (PI) (20 µL/mL) for 10 min. The chondrocytes were then immediately examined by light microscopy and fluorescence microscopy.

#### 2.2.4. LDH 

In the lactate dehydrogenase (LDH) assay, cytotoxicity was determined by the quantification of LDH in the medium. A total of 100 µL of supernatant was added to 100 µL of reagent from the test kit (Cytotoxicity Detection Kit, Roche Molecular Biochemicals, Switzerland) onto a 96-well plate. The activity was determined by colorimetric measurement of the reduction in sodium pyruvate in the presence of NADH. The results were expressed as a percentage of the total enzyme activity expressed by chondrocytes in the presence of clindamycin, gentamicin, and vancomycin.

#### 2.2.5. WST-1

Water-soluble tetrazolium salt (WST-1) was used to determine the metabolic activity of the cells, which is strongly correlated with cell viability. After washing cells in the well plate with PBS twice, the cells were treated for 2 h with the WST-1 solution (triphenyltetrazolium chloride) (WST-1 KIT, Roche Molecular Biochemicals, Switzerland). After this incubation period, formazan solution was formed, which was quantified spectrophotometrically with an ELISA plate reader.

### 2.3. Statistical Analysis

All values were given as mean ± standard deviation. A two-way analysis of variance (ANOVA) Tukey, Fisher’s LSD and Scheffe’s Test were used to evaluate the effect of antibiotic and concentration on cell viability and to compare each antibiotic to each other at each concentration. A *p* value of <0.05 was considered to be significant. Following the preliminary tests, the standard deviation (SD) was hypothesized with 15%. Samples treated with vancomycin reached a survival rate of 70% after 168 h in the preliminary tests. The control reached a survival rate of 90%. The desired statistical power was set at 90%, which determined the sample size of 9 samples needed to achieve an alpha level of 0.025 in a two-sided test. The actual power for an alpha at 0.025 revealed 91.25% using 10 samples. The control revealed a survival rate of 90% (42,630 living cells) and the vancomycin group (10 mg/mL) revealed a survival rate of 72% (10,300 living cells) at an SD of 10% (3000 cells). Statistical analysis was performed with Origin Pro 2023 version 10.0.0.154 (OriginLab, Northampton, MA, USA) and MS Excel 2016 (Microsoft, Redmond, WC, USA).

## 3. Results

Within a 24 h timeframe, notable reductions in cell contacts were observed in chondrocytes subjected to clindamycin and gentamicin, regardless of concentration. The treated cells exhibited smaller sizes and distinct morphological variations compared to the control. Cell extensions, commonly found in monolayer chondrocytes, were minimal or absent. This trend became more pronounced with prolonged exposure, particularly in chondrocytes treated with higher concentrations of antibiotics. After 72 h, a noteworthy reduction in the percentage of living cells became evident in the cells subjected to clindamycin and gentamicin at all concentrations (Table 1). The chondrocytes treated with vancomycin initially showed a more stable cell count until day 3 of the experiment compared to the ones treated with clindamycin and gentamicin. Nevertheless, these cells also exhibited shrinkage and displayed partial disruptions in cell contacts. After 168 h, the control group exhibited the highest number of living cells compared to all other groups. Furthermore, the fluorescence and PI staining of chondrocytes revealed a decline in viable cells (green colouration) as the antibiotic concentrations increased. Additionally, an emergence of necrotic cells (red colouration) was observed with prolonged exposure to the antibiotics (Figure 1). 

After 72 h, LDH activity could be measured in almost all antibiotic-treated cells at all concentrations, whereas chondrocytes treated with vancomycin at both concentrations, and clindamycin at the concentration of 0.5 mg/mL, and gentamicin at the concentration of 5 mg/mL, showed no significant difference in LDH activity after 24 h of exposure, chondrocytes treated with gentamicin at the concentration of 10 mg/mL showed significantly higher LDH activity compared to all other antibiotics (see Figure 2). The chondrocytes treated with clindamycin at 1 mg/mL showed significant differences compared to the other groups after 24 h, but since the LDH release remained below 0%, this can be neglected. After 72 h, there were no significant differences observed in the LDH activities between gentamicin and clindamycin at almost all concentrations, indicating comparable levels of LDH effects induced by gentamicin and clindamycin. However, the LDH activity associated with vancomycin remained consistently lower compared to all other antibiotics and concentrations, demonstrating a distinct and significant variation. This trend persisted until day 7 (168 h), with a continuous increase in LDH activity observed in all antibiotic-treated samples (Figure 3 and Appendix A).

The LDH values were also measured in the positive control, treated with TritonX-100. This control represented the maximum LDH release. The results showed that the LDH levels did not remain constant throughout the 7-day period in the wells (Table 2). With an exponential decrease, this would result in a half-life of LDH in the wells of 6.38 ± 1.53 days. This interpretation of the half-life should be considered with caution, as the nature of the decline (linear or exponential) could not be clearly determined due to the insufficient datasets available over an extended period of time. However, the LDH content decreased, suggesting that a decrease also occurred in the samples themselves. It can be assumed that, on day 7 of the measurement, the entire LDH may not have been secreted. Therefore, the values measured on day 7 do not fully correspond to reality. Consequently, it can be presumed that the antibiotics exhibited an even greater toxicity than indicated by the measurement. 

After 24 h of exposure, formazan levels were measured in all antibiotic-treated samples. The results showed that gentamicin at a concentration of 10 mg/mL exhibited a significantly lower amount of formazan, while vancomycin at a concentration of 5 mg/mL showed a significantly higher amount of formazan compared to the control group. At the 3-day mark, most antibiotics and concentrations demonstrated a significantly lower amount of formazan compared to the control group, except for vancomycin at a concentration of 5 mg/mL, whereas the formazan level remained higher than that of the control group (Figure 4). However, at the 7-day mark, all formazan levels were significantly lower than those observed in the control group (Figure 5).

## 4. Discussion

The primary result of the current study is that all antibiotics tested at different concentrations showed a toxic effect in vitro over the course of 168 h on chondrocytes. However, the three antibiotics examined in this study should be considered in a differentiated manner, as they all showed a similar trend but had varying degrees of impact on chondrocytes. Since there are hardly any studies examining the influence of antibiotics on cartilage cells, analogies from existing studies on the corresponding effect on connective tissue and bone were drawn for discussion. 

According to a study by Roehner et al. [12], vancomycin concentrations below 6.25 mg/mL seldom induced cell necrosis after 24 h. In their investigation, chondrocytes were exposed to vancomycin at various concentrations ranging from 3.125 mg/mL up to 50 mg/mL. The Triphenyltetrazoliumchlorid (XTT) assay conducted by the group over a span of 14 days revealed a significant decrease in the expressed XTT levels on day 5 in chondrocytes exposed to vancomycin at a concentration of 6.25 mg/mL. In line with these results, the concentrations for this study were chosen. Given the concentration at which vancomycin is commonly used in clinical practice, the concentrations for gentamicin and clindamycin were chosen to allow for comparison between the most commonly used antibiotics in orthopaedic surgery [8,17,18,19,20]. Similar results were obtained by Pezzanite et al. [21], who determined an IC50 for vancomycin on chondrocytes of 7.306 mg/mL. In the present study, chondrocytes exhibited a similar behaviour to vancomycin at a concentration of 5 mg/mL in the cell proliferation assay. However, the observations for this study revealed an initial surge in WST-1 expression after 24 h at that concentration. Similarly, Roehner et al.’s [12] results indicated an initial increase in XTT expression, although this only occurred after 48 h. This may be explained by the regenerative capacity of chondrocytes and the toxic threshold of vancomycin at concentrations below 6.25 mg/mL.

In another study conducted by Rathbone et al. [22], the impact of antibiotics belonging to various classes on the viability of osteoblasts and their osteogenic activity was investigated over a 14-day period. The examined antibiotics included gentamicin sulphate and vancomycin. Like the present study, the findings revealed that each antibiotic within a given class led to variations in the extent of the reduction in cell number and, in their case, a decrease in osteogenic activity. Except for vancomycin, all antimicrobial agents demonstrated a cell number retention rate exceeding 50% after 10 days. Notably, vancomycin exhibited no toxic effects until administered at doses of 5 mg/mL. In alignment with the Rathbone study, the present study also substantiated that vancomycin exhibited the least cytotoxic impact on chondrocytes among all the examined antibiotics. Unlike Rathbone’s study, which omitted the initial days of antibiotic exposure, the current study also detected initial cytotoxic effects after only 24 h with clindamycin and gentamicin. However, in contrast to the control, vancomycin exhibited cytotoxic effects only at the higher concentration of 10 mg/mL, starting from day three of treatment. Similar to this finding, Edin et al. [23] reported toxic concentrations of vancomycin at 10 mg/mL on osteoblast-like cells. In vivo, such toxic antibiotic concentrations have not been detected after systemic use. Serum levels of less than 40 µg/mL have been reported [24]. Nevertheless, the local application of vancomycin has been shown to release up to 2000 µg/mL over a period of 12–20 days [25]. In contrast, the XTT expression by chondrocytes exposed to vancomycin at 12.5 mg/mL in the Edins study demonstrates that XTT significantly decreased after 24 h and continued to decline over the duration of the experiment. Similarly, the results of the present study indicated a decrease in WST-1 expression over the course of 7 days. However, in the initial measurement after 24 h, the cells exposed to vancomycin at both concentrations exhibited similar WST-1 expression compared to the control. This trend was also observed with clindamycin at both concentrations. The only statistically significant difference we observed in WST-1 production was after 24 h, between the the control-treated cells and the chondrocytes exposed to vancomycin at 5 mg/mL. This could be a hint that vancomycin, at the concentration of 5 mg/mL, stimulates cell viability after 24 h. However, this effect disappears after 168 h. Naal et al. [26] showed a similar effect of clindamycin on human osteoblasts.

In contrast to our findings, Dogan et al. [27] reported that vancomycin does not exhibit chondrotoxic effects. However, it is important to note that their study utilized very low concentrations of only 0.016 mg/mL. They assessed vital cell number and proliferation rate through a dimethyl thiazole diphenyl tetrazolium salt (MTT) assay, which revealed no cytotoxic effects on chondrocytes. Given the significantly lower concentration employed in their study, these results do not contradict the findings in the present study. It is worth considering that vancomycin is typically locally administered as an antibiotic in much higher concentrations. 

Although it has been shown in spine surgery that the utilisation of locally applied antibiotics led to significant reductions in postoperative infections [28], it was also revealed that the administration of poorly absorbable antibiotics such as vancomycin and gentamicin resulted in high concentrations from hours to days after wound closure [29]. Three pharmacokinetic studies showed that high concentrations of vancomycin could still be detected up to three days after treatment [28,30,31]. Additionally, high concentrations of gentamicin could be found after its local use [32]. 

In the present study, chondrocytes treated with gentamycin were the quickest to show toxic effects in all three investigations. Cells treated at the concentration of 10 mg/mL showed a statistically significant difference in the proliferation assay compared to all other groups. The LDH assay had similar significance, with only clindamycin atthe concentration of 0.5 mg/mL achieving lower results, although these were not statistically significant, regarding LDH release after 24 h. 

While it has been reported that antibiotic-loaded bone grafts substitutes can enhance ontogenesis [31], other studies have shown that gentamicin negatively influenced the osteogenic function [32,33]. This was also found by Chang et al. [34]. In their study, they investigated the toxic effect of gentamicin on stem cells, which were isolated from bone marrow from three adult donors and cultivated with different concentrations of gentamicin, the highest being 200 µg/mL, over the course of 7 days. They tested these cells for type II and X collagen, among other things, and found that, for in vitro chondrogenesis, cells cultivated in ≥100 µg/mL gentamicin expressed significantly less collagen type II and X compared to control-treated cells. This means that, at these concentrations, gentamicin hinders the process of in vitro chondrocyte differentiation. This aligns with our observations from our Live Dead staining assay. Pezzanite et al. [21] report an IC50 value of gentamicin in chondrocytes of 0.708 mg/mL, which is consistent with our observations. The cells in our control and vancomycin-treated groups showed a typical morphological restructuring into fibroblast-like cells; this effect was not observed in the groups cultured with gentamicin or clindamycin. The absence of collagen evidence in the current study prevents us from making a definitive statement about its expression. However, when considering the findings of Chang et al. [33], it is possible that the observed changes in morphology could affect the expression of collagen, thereby limiting chondrogenesis in the presence of gentamicin. The impact of aminoglycoside antibiotics on bone-healing functions and the differentiation pathway remains uncertain to date [34]. In the present trial, it could not be confirmed whether gentamicin had a positive effect on chondrocytes. However, it should also be taken into consideration that the concentrations used in the present study were much higher than those described in other previous studies.

The change in morphology, as well as the decrease in cell number, the rapid release of LDH and the decrease in WST-1 expression, lead to the conclusion that gentamicin exerts a clear toxic effect on chondrocytes, as described previously for other cell lines [22,34,35]. The same change in behaviour could be observed in the present study in chondrocytes treated with clindamycin, and to a lesser extent in cells treated with vancomycin. 

The antibiotic clindamycin is commonly used in topical medications to treat osteomyelitis and infections associated with implantable materials in orthopaedic surgery. Although this antibiotic is widely used and local concentrations were reported to exceed 1000 µg/mL [36,37], there is still a lack of information on the direct effects of clindamycin on the cell function of chondrocytes. As for vancomycin and gentamycin, studies have shown a cytotoxic effect of clindamycin on osteoblasts [26]. Naal et al. [26] reported osteoblastic proliferation at the lowest tested clindamycin concentration of 10 µg/mL and an increased LDH level after cell exposure to clindamycin concentrations of 0.5 mg/mL after 48 and 72 h. Additionally, they demonstrated that at a lower concentration of 0.01 mg/mL, clindamycin stimulated the cell metabolism of human chondrocytes. Although no increase in metabolism was detected in the present study, which may be explained by the high concentrations of clindamycin used, the results are consistent with those of Naal et al. [26]. After 72 h, more than 40% of the chondrocytes treated with 0.5 mg/mL clindamycin released LDH, whereas after 24 h of the experiment, they did not release LDH at all. Similar results could be shown at the even higher concentration of 1 mg/mL, where the release of LDH was at almost 60% after 72 h. 

Limitations: The current study did not replicate an in vivo environment. In vitro experiments may exhibit higher cellular toxicity when chondrocytes are directly exposed without the protective presence of a three-dimensional extracellular matrix. The antibiotics examined in vitro do not encounter barriers like the cartilage matrix. Additionally, our investigation involved cultured chondrocytes that were already in a differentiated state, potentially rendering them more susceptible to toxic substances.

## 5. Conclusions

The current study has revealed the detrimental effects of clindamycin, gentamicin and vancomycin on the proliferation and metabolism of human chondrocytes in vitro. Of these, vancomycin showed the lowest cytotoxic effect. Further research is required to assess the impact of elevated local antibiotic concentrations in vivo on local cartilage metabolism. 

## Figures and Tables

**Figure 1 biomedicines-11-03143-f001:**
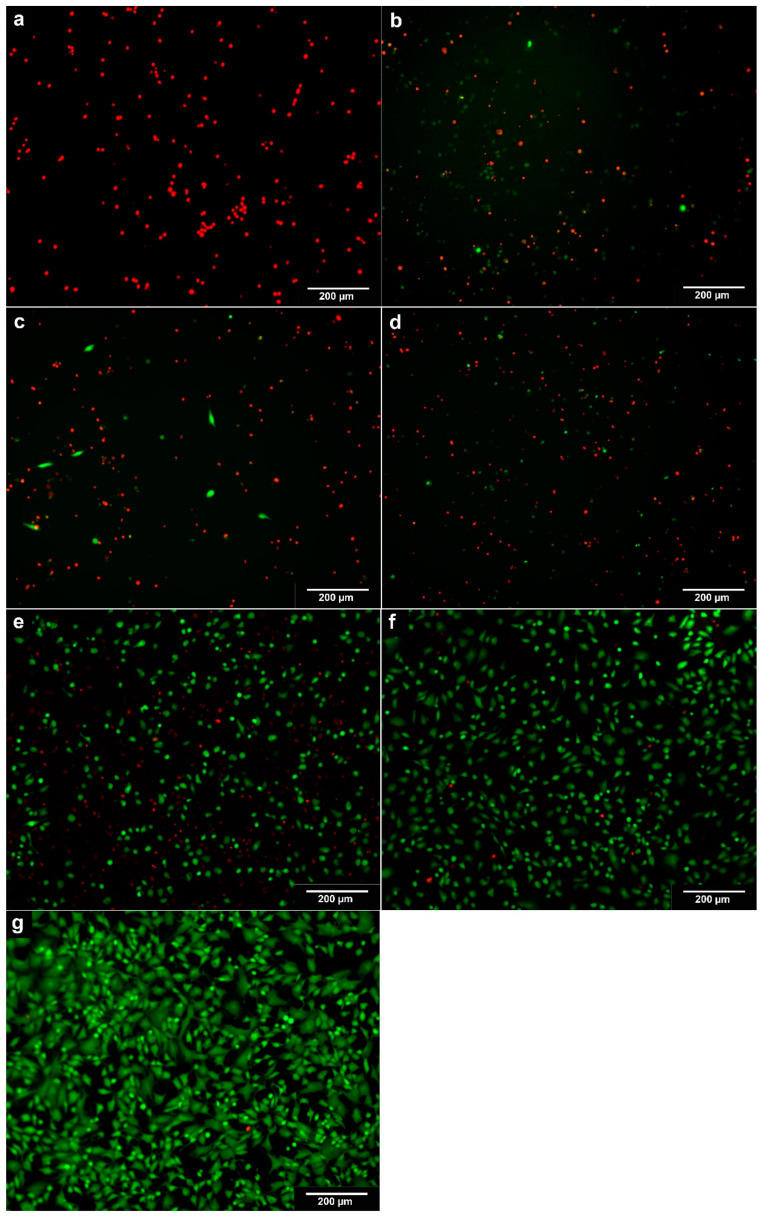
PI and calcein staining of chondrocytes after 168 h; (**a**): Clindamycin 1 mg/mL, (**b**): Clindamycin 0.5 mg/mL, (**c**): Gentamicin 10 mg/mL, (**d**): Gentamicin 5 mg/mL, (**e**): Vancomycin 10 mg/mL, (**f**): Vancomycin 5 mg/mL, (**g**): control; images taken with Carl ZEISS Apotome.2, 10x magnification; living cells = green; apoptotic cells = red.

**Figure 2 biomedicines-11-03143-f002:**
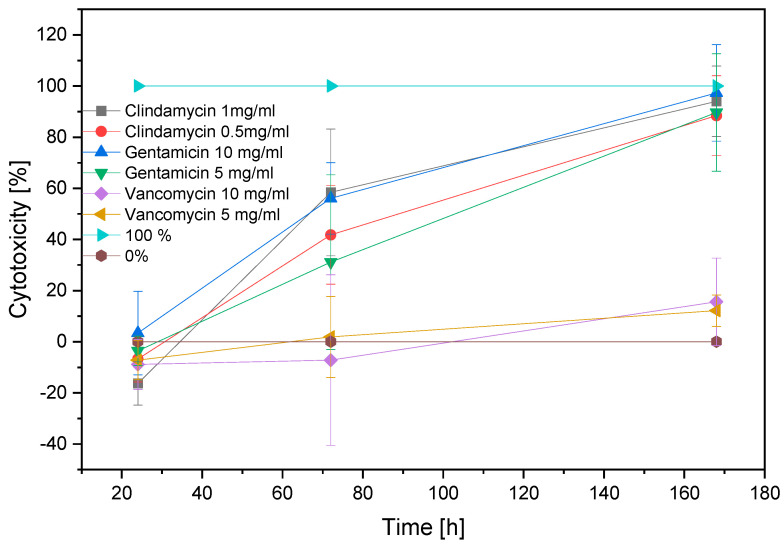
Cytotoxicity over time 24–168 h; 0% (negative control for apoptotic cells); 100% (TritonX-treated cells, positive control for apoptotic cells).

**Figure 3 biomedicines-11-03143-f003:**
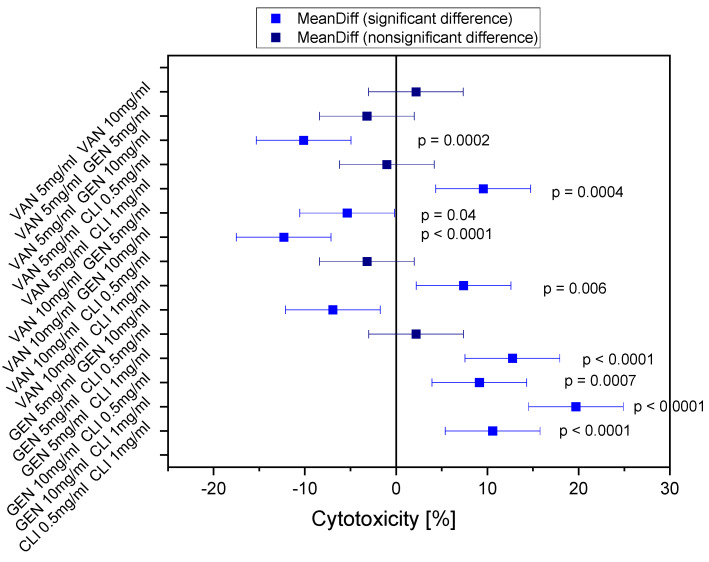
Significances in cytotoxicity after 24 h; Vancomycin (VAN); Clindamycin (CLI); Gentamicin (GEN); significant difference = blue; non-significant difference = black).

**Figure 4 biomedicines-11-03143-f004:**
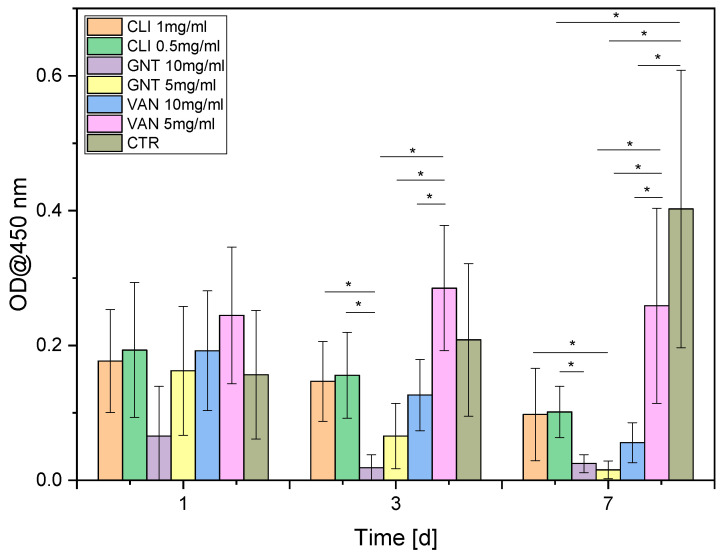
WST-1 measurement over time 1–7 d; Clindamycin (CLI); Gentamycin (GNT), Vancomycin (VAN); control (CTR); *p* < 0.05 (*).

**Figure 5 biomedicines-11-03143-f005:**
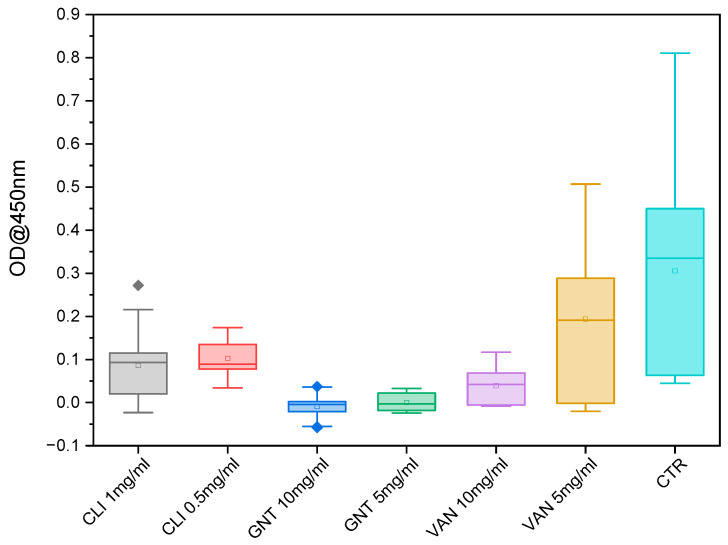
Box Plot WST-1 measurement after 7 d; CLI (Clindamycin (CLI); Gentamycin (GNT), Vancomycin (VAN); control (CTR).

**Table 1 biomedicines-11-03143-t001:** Percentage of live cells depending on observation time.

Living Cells [%]							
Time [h]	Control	Clindamycin 1 mg/mL	Clindamycin 0.5 mg/mL	Gentamicin 10 mg/mL	Gentamicin 5 mg/mL	Vancomycin 10 mg/mL	Vancomycin 5 mg/mL
24	96 ± 3	66 ± 20	79 ± 16	89 ± 5	92 ± 4	93 ± 4	93 ± 4
72	96 ± 3	14 ± 11	54 ± 20	42 ± 10	58 ± 16	91 ± 5	93 ± 3
168	90 ± 13	0	9 ± 7	8 ± 8	11 ± 11	72 ± 19	75 ± 22

**Table 2 biomedicines-11-03143-t002:** LDH release in positive control group.

Time in Days	LDH Release in Positive Control [16]
1	1.51 ± 0.42
2	1.33 ± 0.09
3	1.05 ± 0.19
7	0.81 ± 0.17

## Data Availability

The data presented in this study are available on request from the corresponding author.

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
