# Peer review of "Effect of Vancomycin, Gentamicin and Clindamycin on Cartilage Cells In Vitro"

_biomedicines, 2023, doi:10.3390/biomedicines11123143_

Round 1

Reviewer 1 Report

Comments and Suggestions for Authors

Authors have minor revision:

Line 56. sports traumatology – please delete it (there is sports medicine not sports traumatology or sports orthopaedics).

Line 141. Enzyme activity –  correct to „enzyme activity“

Line 271. decrease space between two sentence.

Paper is written very good with topic that is very interested for orthopaedic surgeons worldwide. Use of topic antibiotics for decreasing infection rate in orthopaedic surgery is hot topic. Much more laboratory studies are necessary to find a best solution.

Author Response

Dear Reviewer 1,

Thank you very much for taking the time to review our manuscript. Our answers to your questions / suggestions are listed below and colored blue for a better overview.

Authors have minor revision:

Line 56. sports traumatology – please delete it (there is sports medicine not sports traumatology or sports orthopaedics).

Thank you, we changed the sentence into “… sports medicine..”

Line 141. Enzyme activity –  correct to „enzyme activity“

We removed the capital letter and now it´s “enzyme activity”

Line 271. decrease space between two sentence.

We removed the space between the two sentences in Line 271 (after revision 269)

Paper is written very good with topic that is very interested for orthopaedic surgeons worldwide. Use of topic antibiotics for decreasing infection rate in orthopaedic surgery is hot topic. Much more laboratory studies are necessary to find a best solution.

Best regards 

the authors

Reviewer 2 Report

Comments and Suggestions for Authors

Manuscript: Effect of Vancomycin, Gentamicin and Clindamycin on cartilage cells in vitro

The rational of the study is unclear. There are a few concerns with the hypothesis and study design. 

For instance, authors states "Given the concentration at which vancomycin is commonly used in clinical practice, the concentrations for gentamicin and clindamycin were chosen to allow comparisons between the most used antibiotics in orthopaedic surgery" selecting "In vitro" concentration of two antibiotics based on third's clinically practiced dose is irrational. In vitro and in vivo effective concentrations hugely vary from drug to drug. 

The effects of antibiotics are tested in vitro on chondrocyte? What was the passage number of cells used for the experiments? 

Did authors pulled the chondrocytes from all 10 patients to perform the experiments? Or the experiments were performed as 10 independent biological replicates? 

The manuscript is very poorly written. In Figure 2 the text box is over the data lines. Figure legends does not correspond to the discussion (See Fig.3 24hours vs Line #197 day07).

Figure 2: 100%.....TritinX. What does that mean? no explanation in the figure legends or methods. 

Figure 2 and figure 4 graphs lack any significance. 

Line #354 and #355 is unclear. The introduction is primarily focused on tendon/ligament whereas the discussion bone graft in discussion section where as conclusion mention bone metabolism. The manuscript needs extensive revision for a proper flow in the text. 

Authors need to provide catalog number for the kits and chemicals/drugs used. 

Comments on the Quality of English Language

Grammatical errors. Days and hours time duration is not used uniformly.  

Author Response

Dear Reviewer 2,

Thank you for taking the time to review our manuscript. Our answers to your questions / suggestions are listed below and colored blue for a better overview.

The rational of the study is unclear. There are a few concerns with the hypothesis and study design.

The rational including the primary endpoint are given in Lines 66-70: “…The aim of the current trial is to investigate the toxicity towards cartilage cells of clindamycin, gentamicin and vancomycin over the course of up to 7 days. The primary endpoint is the assessment of the relative survival rate of chondrocytes following a 168-hour exposure to antibiotics at various concentrations, in comparison to a control group that does not receive antibiotic treatment. …”

For instance, authors states "Given the concentration at which vancomycin is commonly used in clinical practice, the concentrations for gentamicin and clindamycin were chosen to allow comparisons between the most used antibiotics in orthopedic surgery" selecting "In vitro" concentration of two antibiotics based on third's clinically practiced dose is irrational. In vitro and in vivo effective concentrations hugely vary from drug to drug.

We overworked the Abstract: “… In vitro and in vivo effective concentrations hugely vary from drug to drug. Given the concentration at which vancomycin is commonly used in clinical practice, the concentrations for gentamicin and clindamycin were chosen from literature [5-9] to allow comparisons between the most used antibiotics and its concentration range in orthopedic surgery …”

We chose these sources because we wanted to cover a wide range and because not much has been published on this subject. But you are right, of course this varies a lot, especially since it depends on the surgeon - but we had to commit ourselves.

The effects of antibiotics are tested in vitro on chondrocyte? What was the passage number of cells used for the experiments?

The cells were not passaged - we harvested the cells from the patients' cartilage tissue and seeded them. Once the cell culture flasks were 80% confluent harvested and used.

Did authors pooled the chondrocytes from all 10 patients to perform the experiments? Or the experiments were performed as 10 independent biological replicates?

We performed the experiments on 10 independent biological specimen. The cells were not pooled.

The manuscript is very poorly written. In Figure 2 the text box is over the data lines. Figure legends does not correspond to the discussion (See Fig.3 24hours vs Line #297 day07).

You are right the data box in Fig. 2 is above the 100% line - but it only covers the first value of the positive control (100% dead on the TritonX), so we haven't really seen any problems there, especially since we always design it that way.

You are right, we have changed the value in the discussion, it should be 0.5 instead of 1.

Figure 2: 100%.....TritinX. What does that mean? no explanation in the figure legends or methods.

TritonX 100% is the positive control. 100% apoptotic cells after the addition of the TritonX. You are right, we have added to the caption accordingly and to Mat&Meth

Figure 2 and figure 4 graphs lack any significance.

that was not the goal at all - we only wanted to show the course of proliferation as well as cytotoxicity. These are common diagrams, as shown in many other publications (especially in biocompatibility tests).

Line #354 and #355 is unclear. The introduction is primarily focused on tendon/ligament whereas the discussion bone graft in discussion section where as conclusion mention bone metabolism. The manuscript needs extensive revision for a proper flow in the text.

Since there are hardly any studies examining the influence of antibiotics on cartilage cells, analogies from existing studies on the corresponding effect on connective tissue and bone were drawn for the discussion.

We noted this at the beginning of the discussion

Authors need to provide catalog number for the kits and chemicals/drugs used.

You are right - we have added this at the appropriate places

Comments on the Quality of English Language

Grammatical errors. Days and hours time duration is not used uniformly. 

A native speaker of the institution (Assoc. Prof, PHD, Biologist, USA) did a final proofread and language correction of the paper. 

Round 2

Reviewer 2 Report

Comments and Suggestions for Authors

Figure 2 and figure 4 graphs lack any significance.

that was not the goal at all - we only wanted to show the course of proliferation as well as cytotoxicity. These are common diagrams, as shown in many other publications (especially in biocompatibility tests).

Significance here is referred as the p-values (Statistical significance between data points) as represented in Fig.3.

Figure - 2 still has label box ON the lines of the graph and if the data is not significant then authors should remove that data point.  

Author Response

Dear Rviewer 2,

Thank you for taking the time to review our arcticle. Our Answers were highlighted in blue:

Figure 2 and figure 4 graphs lack any significance.

that was not the goal at all - we only wanted to show the course of proliferation as well as cytotoxicity. These are common diagrams, as shown in many other publications (especially in biocompatibility tests).

We are aware of this - but in the case of cytotoxicity, only the cytotoxicities of the individual samples should be shown and NOT compared. The line-dot diagram is not suitable for this purpose. But you are right the significances in Fig. 4 has been added.

Significance here is referred as the p-values (Statistical significance between data points) as represented in Fig.3.

That's exactly why we added figure 3 in the first place ;) to show the statistical significance between the data points. We added the significances for the other days to the supplement (Figure S1)

Figure - 2 still has label box ON the lines of the graph and if the data is not significant then authors should remove that data point.  

We have moved the box

Best regards

the authors